# MiR-302 Regulates Glycolysis to Control Cell-Cycle during Neural Tube Closure

**DOI:** 10.3390/ijms21207534

**Published:** 2020-10-13

**Authors:** Rachel A. Keuls, Karin Kojima, Brittney Lozzi, John W. Steele, Qiuying Chen, Steven S. Gross, Richard H. Finnell, Ronald J. Parchem

**Affiliations:** 1Development, Disease Models & Therapeutics Graduate Program, Baylor College of Medicine, One Baylor Plaza, Houston, TX 77030, USA; Rachel.Keuls@bcm.edu; 2Center for Cell and Gene Therapy, Stem Cells and Regenerative Medicine Center, One Baylor Plaza, Houston, TX 77030, USA; Karin.Kojima@bcm.edu; 3Genetics and Genomics Graduate Program, Baylor College of Medicine, One Baylor Plaza, Houston, TX 77030, USA; Brittney.Lozzi@bcm.edu; 4Department of Molecular and Cellular Biology, Baylor College of Medicine, Houston, TX 77030, USA; John.Steele@bcm.edu (J.W.S.); Richard.Finnell@bcm.edu (R.H.F.); 5Center for Precision Environmental Health, Department of Molecular and Cellular Biology and Medicine, Baylor College of Medicine, Houston, TX 77030, USA; 6Department of Pharmacology, Weill Cornell Medicine, New York, NY 10065, USA; qic2005@med.cornell.edu (Q.C.); ssgross@med.cornell.edu (S.S.G.)

**Keywords:** microRNA, *miR-302*, proliferation, neural tube closure, neural tube defects, glycolysis, maternal diabetes, hyperglycemia, oxidative stress, cell cycle

## Abstract

Neural tube closure is a critical early step in central nervous system development that requires precise control of metabolism to ensure proper cellular proliferation and differentiation. Dysregulation of glucose metabolism during pregnancy has been associated with neural tube closure defects (NTDs) in humans suggesting that the developing neuroepithelium is particularly sensitive to metabolic changes. However, it remains unclear how metabolic pathways are regulated during neurulation. Here, we used single-cell mRNA-sequencing to analyze expression of genes involved in metabolism of carbon, fats, vitamins, and antioxidants during neurulation in mice and identify a coupling of glycolysis and cellular proliferation to ensure proper neural tube closure. Using loss of *miR-302* as a genetic model of cranial NTD, we identify misregulated metabolic pathways and find a significant upregulation of glycolysis genes in embryos with NTD. These findings were validated using mass spectrometry-based metabolite profiling, which identified increased glycolytic and decreased lipid metabolites, consistent with a rewiring of central carbon traffic following loss of *miR-302*. Predicted *miR-302* targets *Pfkp*, *Pfkfb3*, and *Hk1* are significantly upregulated upon NTD resulting in increased glycolytic flux, a shortened cell cycle, and increased proliferation. Our findings establish a critical role for *miR-302* in coordinating the metabolic landscape of neural tube closure.

## 1. Introduction

Neural tube closure is a critical first step in the formation of the central nervous system and requires precise cell growth, movement, and organization [1,2,3,4,5,6,7,8,9,10,11,12,13]. Neural tube closure begins early in embryonic development near the end of gastrulation when the bi-lateral halves of the neuroepithelium rise to form neural folds that converge to form a closed neural tube. Neural tube closure defects (NTDs) occur in approximately 1 in every 1000 live births, of which anterior closure defects are the most severe with the highest incidence of lethality [14,15]. There is a lack of tools to diagnose and intervene in the early stages of pregnancy, leading to lifelong disability in patients with NTDs [16,17]. The causes of NTDs are heterogeneous and have been shown to result from an interaction of both environmental and genetic factors [8,9,10,14,17,18,19,20,21,22,23].

In particular, NTDs in humans have been associated with changes in the oxidative state of cells within the developing neural tube resulting from exposure to teratogens, such as excess glucose in the context of maternal diabetes [24,25,26,27,28,29]. Indeed, women who have a high dietary glycemic load are more likely to have an offspring with an NTD [30,31,32,33,34]. Diabetic mouse models have revealed that antioxidant genes are downregulated further increasing the oxidative state of the cell [35]. How increased glucose and oxidative state changes affect fetal metabolism and increase NTD risk has not been fully elucidated.

During neural tube closure, before the formation of the placenta and a functional fetal circulatory system, the cellular environment is hypoxic, and ATP is generated largely through glycolysis [36,37,38]. Glycolysis provides not only ATP but also metabolic intermediates that are essential for cell growth, proliferation, and differentiation, including nucleotides, amino acids, and fatty acids. Quiescent, differentiated cells utilize low levels of glucose and predominantly oxidize glucose carbons to carbon dioxide via the TCA cycle [39]. In contrast, highly proliferative undifferentiated stem cells import and metabolize vast amounts of glucose, exhibiting a very rapid glycolytic flux [40,41]. Increased proliferation of the neuroepithelium during neural tube closure results in exencephaly, and, therefore, proliferation must be tightly regulated [42,43]. However, the link between glycolysis and the control of proliferation during neural tube closure remains understudied.

The unique cell cycle and self-renewal properties of human embryonic stem cells (hESCs) are maintained by a set of microRNAs (miRNAs), 21–23 nucleotide long single-stranded RNAs that repress gene expression in most animal species [44,45,46,47,48,49]. *MIR-302* family members are among the most highly expressed miRNAs in hESCs and play a critical role in pluripotent potential [49,50,51,52,53]. Deletion of *miR-302* in mice results in a fully penetrant NTD and an increase in proliferation of neuroepithelial cells [43]. Notably, excessive cell proliferation has been previously associated with NTDs [54]. How *miR-302* controls cell proliferation during neural tube closure has not been thoroughly investigated.

Here, we used single-cell mRNA sequencing and mass-spectrometry to compare the metabolic profile between successful and failed neurulation in mice. Using the *miR-302* knockout mouse model to induce a cranial neural tube closure defect, we identified an upregulation of genes involved in glycolysis including predicted *miR-302* targets *Pfkp*, *Pfkfb3*, and *Hk1* which were upregulated 2-4.5-fold. Metabolic profiling confirmed that intermediates produced by predicted *miR-302* targets are increased upon loss of *miR-302*. Further differential expression analysis revealed an enrichment for genes that promote cellular proliferation in cells with upregulated *Pfkp*, *Pfkfb3*, or *Hk1*. In *miR-302* knockout ectoderm, we found upregulation of Pfkfp3 which promotes expression of Cdk1 and Cdk4. We observed an increase in the number of cells co-expressing G2/M and G1/S genes—consistent with accelerated proliferation of the *miR-302* knockout ectoderm. Taken together, our results implicate *miR-302* in the control of the upper glycolysis pathway carbon flux to regulate proliferation of the ectoderm during neural tube closure.

## 2. Results

### 2.1. Metabolism and Differentiation Are Coupled During Neurulation

To analyze gene expression during neurulation, we used single-cell mRNA sequencing of the cranial region from E8.25 and E9.5 mouse embryos. Seurat was used to cluster cells into populations which were identified similarly to previous studies [55] (Appendix A). We focused specifically on ectoderm-derived cell populations that contribute to neural tube closure, including fore-, mid-, and hind-brain tissues of the central nervous system, the neural crest lineage, and non-neural ectoderm (Figure 1A). To identify ectoderm-specific genes which may be critical within the developing neural tube, we used an unbiased differential expression analysis to compare ectoderm-derived cell populations with non-ectoderm-derived cells during neural tube closure at E8.25 (Figure 1A–C and Appendix A) and E9.5 (Figure 2D–F and Appendix A). At the initiation of neural tube closure at E8.25, differential expression analysis revealed an upregulation of genes within ectoderm-derived cells involved in glucose import such as *Slc2a1* and *Slc2a3*, as well as stem cell transcription factors including *Sox2* and *Sall4* (Figure 2B) [56,57]. Genes encoding mitochondrial membrane proteins and ATP/ADP transporter proteins such as Slc16a1 and Slc25a4 were similarly enriched (Figure 1B). Gene ontology (GO) of genes enriched in ectoderm versus other embryonic cell types at E8.25 revealed predicted roles in metabolic activity, actin filament-, cadherin-, and retinoid-binding, as well as cell adhesion, consistent with differentiation and morphogenesis of the neural tube (Figure 1C).

The same unbiased differential expression analysis was applied to E9.5 embryos, comparing ectoderm-derived versus non-ectoderm-derived cells after completion of neural tube closure (Figure 1D). Similar to our findings at E8.25, metabolic genes such as the glucose transporter *Slc2a3* and stem cell factor *Sox2* were among the differentially expressed genes at E9.5 (Figure 1E). In contrast to E8.25, we found an upregulation of genes at E9.5 that promote differentiation of the neuroepithelium (Figure 1E,F), such as Fgf15 [43]. We identified a slight downregulation of antioxidant *Selenop* and the glutathione peroxidase, *Gpx1,* suggesting a decrease in antioxidant activity during neural tube closure (Figure 1E). Consistent with a decline in antioxidant activity, *Mt1*, an antioxidant gene activated by the redox transcription factor *Nrf1* [58], was downregulated in the ectoderm at E9.5 (Figure 1E). GO analysis of differentially expressed genes at E9.5 revealed similar predicted functions at E8.25, including categories such as glucose transmembrane transport and the cell–cell adhesion involved in the morphogenesis of the neural tube. (Figure 1F). The results of differential expression analysis revealed an enrichment of genes that are involved in metabolism and genes that regulate cell differentiation while GO analysis revealed terms associated with both metabolism and morphogenesis of the neural tube. Together, these results suggest a link between gene programs involved in glucose metabolism and differentiation within the ectoderm during the progression of neural tube closure.

### 2.2. Single-Cell Sequencing Reveals Metabolic Contributions to Neural Tube Closure

Since we found an enrichment for metabolism genes in the developing neural tube and NTDs result from changes in energy metabolism and redox state of the cell, we analyzed how the expression of genes involved in fatty acid metabolism, antioxidant activity, oxidative phosphorylation, glycolysis, and folic acid metabolism change over time during neural tube closure in a cell and tissue-specific manner. We again focused on ectoderm-derived cell populations that contribute to neural tube closure (Figure 2A). Our analysis of metabolic gene expression in the ectoderm found that genes involved in oxidative phosphorylation and glycolysis were more highly expressed than genes involved in fatty acid metabolism, antioxidant activity, and folic acid metabolism at both E8.25 and E9.5 (Figure 2B,C).

To analyze differences in metabolic gene expression between each of the ectoderm-derived cell populations, we normalized gene expression in each population to the lowest expression level within each metabolic category. Genes involved in glycolysis and folic acid metabolism were found to vary more in expression between the ectoderm-derived cell populations, as compared to genes involved in fatty acid metabolism, antioxidant activity, or oxidative phosphorylation (Appendix A). For example, the expression of glycolysis genes was 1.2-fold greater in the neuroepithelium and neural crest relative to non-neural ectoderm at E8.25 (Appendix A). Interestingly, folic acid metabolism genes were expressed 1.3-fold higher in the neural crest versus the neuroepithelium at E9.5, suggesting that neural crest cells actively process folate during neural tube closure (Appendix A). Folic acid has previously been associated with promoting neural crest differentiation and outgrowth of neuroepithelial cells [59].

Consistent with the enrichment for glycolysis genes in the developing neural tube, *Hif1a* and *Nrf1*, two redox-sensing transcription factors that regulate the expression of glycolysis genes [60,61], were enriched in ectoderm-derived cells during neural tube closure (Appendix A). Notably, redox sensing transcription factors activate the expression of antioxidant enzymes which were also shown to be enriched in the ectoderm at E8.25, the earliest stage in neural tube closure (Appendix A).

To summarize the changes in metabolic gene expression during neural tube closure, we compared E8.25 to E9.5 and observed a progressive increase in expression of folic acid genes during neural tube closure, consistent with the essential requirement for folate. Corroborating the enrichment of folic acid metabolism genes in the neural crest, we found a greater increase in folic acid gene expression in the neural crest versus the neural tube (Figure 2D,E). High expression of glycolysis and oxidative phosphorylation genes were maintained throughout the ectoderm until neural tube closure, while expression of genes involved in antioxidant activity decreased.

### 2.3. Upper Glycolysis Is Regulated by miR-302

To identify how metabolism may be altered in embryos with a cranial NTD, we used single-cell sequencing of the cranial region of the E9.5 *miR-302* knockout embryo. The loss of *miR-302* leads to a fully penetrant NTD with excessive proliferation of the neuroepithelium and precocious neuronal differentiation (Figure 3A,B) [43]. Similar to the rapid proliferation of glycolytic embryonic stem cells, we found that genes involved in glycolysis were upregulated across all ectoderm-derived cells in the *miR-302* knockout embryo as compared to the wildtype embryo, correlating with increased proliferation of the neuroepithelium (Figure 3C–E and Appendix A) [40,41,43]. As anaerobic glycolysis is known to be activated by the redox transcription factor *Hif1a* under hypoxic conditions [60], we examined if *Hif1a* expression was enhanced following deletion of *miR-302* and found no evidence of altered *Hif1a* gene regulation (Appendix A). In contrast to glycolysis, we found that genes involved in antioxidant activity, oxidative phosphorylation, and folic acid metabolism were downregulated in the *miR-302* knockout (Figure 3C–E). Furthermore, our data reveal the downregulation of folic acid metabolism genes to be the most drastically impacted by deletion of *miR-302*. This result suggests that a deficiency in folate processing may contribute to the NTD of *miR-302* knockout embryos (Figure 3C–E).

To further dissect the upregulation of the glycolysis pathway in *miR-302* knockout embryos, we analyzed misregulation of genes involved in glucose import/phosphorylation, upper glycolysis, lower glycolysis, the metabolic shift from glycolysis to oxidative phosphorylation, gluconeogenesis, and the pentose phosphate pathway (Figure 3F–H). Ectoderm-derived cells of the *miR-302* knockout embryo had a significant upregulation of glucose import/phosphorylation and upper glycolysis genes, as compared to more downstream glycolysis genes, the expression of which was minimally misregulated (1–1.2 fold change) (Figure 3F–H). These data suggest that *miR-302* may restrain upper glycolysis during neural tube closure and the loss of this post-transcriptional regulatory pathway may contribute to the NTD (Figure 3E–H and Appendix A).

### 2.4. Loss of miR-302 Leads to Accumulation of Glycolytic Intermediates

To identify misregulated metabolites upon *miR-302* deletion, we used mass spectrometry to profile the cranial region of wildtype and *miR-302* knockout embryos at E9.5 (Figure 4A,B and Appendix A). Untargeted metabolomic analysis revealed a general increase in glycolytic metabolites of the *miR-302* knockout embryo, whereas minimal changes in antioxidant and vitamin metabolites were observed. Surprisingly, metabolites associated with fatty acid synthesis and oxidation were also found to be downregulated in the *miR-302* knockout suggesting that carbons may be shuttled into glycolysis away from the fatty acid synthesis pathway (Figure 4C–E and Appendix A). Thirty percent (3 of 10) of metabolites associated with glycolysis were significantly misregulated upon *miR-302* deletion. This resulted in a cumulative 1.6-fold increase in total glycolytic metabolites. Specifically, D-fructose 1/2,6-bisphosphate was among the top significantly upregulated metabolites, with a 1.9-fold increase following *miR-302* deletion (Figure 4C,D). Interestingly, the top upregulated metabolite was docosapentaenoic acid (2.2-fold) which is a long chain omega-3 fatty acid. Docosapentaenoic acid is known to promote brain development and function, and embryos which are deficient in Omega-3 have reduced brain size [42]. Thus, upregulation of long chain fatty acids may contribute to increased proliferation of *miR-302* knockout embryos [43]. Additionally, we analyzed differences in pyruvate and ATP and found a 1.4- and 1.8-fold upregulation, respectively, consistent with increased glycolytic flux-mediated energy generation (Figure 4F,G).

### 2.5. MiR-302 Targets Pfkp, Pfkfb3, and Hk1 to Regulate Upper Glycolysis

To analyze how *miR-302* may be regulating glycolysis, we identified 12 predicted *miR-302* targets that were among the 67 glycolysis genes analyzed and found that 4 of the 12 targets are part of the upper glycolysis pathway (Figure 5A and Appendix A). Of the 12 predicted targets, three were significantly upregulated across all ectoderm-derived cell populations upon *miR-302* deletion; *Hk1* which catalyzes glucose phosphorylation, the first enzymatic step of glycolysis and inhibits apoptosis [62,63,64]; *Pfkp* which catalyzes the conversion of fructose 6-phosphate to fructose 1,6-bisphosphate [65] and known to promote cell growth and proliferation [66,67]; and *Pfkfb3,* an isoform of the 6-phosphofructo-2-kinase/fructose-2,6-bisphosphatases which catalyzes the formation of fructose 2,6-bisphosphate [68] for accelerated glycolytic flux [69] (Figure 5B and Appendix A). Notably, *Pfkp* was upregulated 4.5-fold across all ectoderm-derived cell populations, *Pfkfb3* was upregulated 2.5-fold, and *Hk1* was upregulated 2-fold (Appendix A). Mass spectrometry revealed fructose 1/2,6-bisphosphate to be the only significantly upregulated glycolytic metabolite with a fold-change greater than 1.5 (Figure 5C and Appendix A). To validate the upregulation of Pfkp, we isolated protein from three wildtype and three *miR-302* knockout embryos and measured absorbance during the conversion of fructose 6-phosphate to fructose 1/2,6-bisphosphate. We found a significantly increased rate of fructose 1/2,6-bisphosphate formation of the *miR-302* knockout embryo (Figure 5D–G). *Pfkfb3* was markedly upregulated in the hindbrain, the region of the neural tube that is consistently found to have a closure defect in the *miR-302* knockout model. *Pfkfb3* produces fructose 2,6-bisphosphate, which serves as an allosteric activator of *Pfkp,* thus allowing the bypass of *Pfkp* inhibition by an accumulation of unused ATP within the cell, thus further elevating glycolytic flux (Appendix A).

### 2.6. miR-302 Targets Pfkp, Pfkfb3, and Hk1 to Regulate Cell Proliferation

To investigate the effect of an upregulation of *miR-302* targets within upper glycolysis on neural tube closure, we used differential expression to identify genes significantly and similarly enriched in cells with upregulated *Pfkp*, *Pfkfb3,* or *Hk1* (fold change >1). Interestingly, this analysis identified enrichment of *miR-302* targets that are predicted to have an important role during neurulation (Figure 6A and Appendix A). For example, GO analysis of differentially expressed genes was enriched for terms such as cell proliferation, migration, and differentiation, consistent with a link between glycolysis and cell proliferation being regulated by *miR-302* (Figure 5B). Furthermore, *Fgf15* was upregulated consistent with precocious differentiation, and *Top2a* was upregulated consistent with an increased rate of cell proliferation (Figure 6A) [43,70].

To further investigate the potential link between cell cycle and metabolism, we analyzed expression of cell cycle checkpoint regulators such as Cyclins and Cyclin-dependent-kinases (Cdks). We hypothesized that proliferation of the neuroepithelium of *miR-302* knockout embryos may give rise to NTDs by shortening of the cell cycle. A more rapid cell cycle could occur due to increased expression of genes promoting G2/M and G1/S transitions or if the expression of transition genes is prolonged such that G2/M and G1/S genes are co-expressed for an abbreviated G1 phase and more rapid proliferation [71]. We found that most G2/M or G1/S genes were minimally misregulated following deletion of *miR-302* (Figure 6C and Appendix A). We also analyzed the expression of Cdk and Cyclin mRNA across all ectoderm populations and found minimal changes upon *miR-302* deletion (Figure 6D). Next, we asked whether co-expression of G2/M and G1/S genes could be associated with increased proliferation. Interestingly, co-expression (expression of both genes >1) of G1/S and G2/M genes was increased in the *miR-302* knockout ~1.6-fold. Interestingly, further analysis of cells that co-express the G1/S cyclin (*CyclinD1*) and the G2/M cyclin (*CyclinB1*) revealed an upregulation of cyclin mRNA upon *miR-302* deletion (Figure 6E). Previous studies found that the product of PFKFB3 increases the expression and activity of CDK1 [72] and PFKFB3 directly binds to CDK4 to promote cell cycle progression [35]. We found an increase in the number of ectoderm-derived cells co-expressing *Pfkfb3* and *Cdk1 or Cdk4* (expression of both genes >1) upon *miR-302* deletion (Figure 6F). These data suggest that *miR-302* selectively regulates mRNA encoding enzymes within the upper glycolysis pathway that control synthesis of metabolites which interact with checkpoint regulators for cell cycle regulation.

Taken together, our results suggest that *miR-302* deletion significantly impacts glycolysis and folic acid metabolism, processes that must be tightly regulated for proper neural tube closure. We find glycolysis to be coupled with differentiation and proliferation within the developing neural tube. Our model suggests that *miR-302* targets *Pfkp*, *Pfkfb3*, and *Hk1* within upper glycolysis to cooperatively restrain the level of glycolytic flux, impacting proliferation in ectoderm-derived cells (Figure 6G). Misregulation of these pathways may contribute to the development of human NTDs. Our findings reveal the critical importance of glycolytic flux, energy generation, and their interaction with cell cycle regulators during developmental progression of neural tube closure.

## 3. Discussion

Using unbiased single-cell RNA and metabolite profiling, we show that glycolysis genes and pathway intermediates are maintained at high levels during neural tube closure. NTDs resulting from loss of *miR-302* were associated with upregulation of glycolysis genes and a downregulation of folic acid genes, alterations that have both been shown to promote NTDs [30,31,32,33,34,73]. Previous studies have identified glycolysis as the primary source of energy during neural tube closure at E8.5 [74]. Our work demonstrates a role for post-transcriptional regulation of glycolysis in specific cells of the neural tube using single-cell transcriptomic analysis. Expression of *miR-302* during embryonic development is consistent with an important role in regulating glycolysis. At implantation, the primary source of ATP switches from oxidative phosphorylation to glycolysis [75,76,77], and *miR-302* becomes highly expressed in the post-implantation epiblast [78]. Thus, glycolysis and *miR-302* expression are coupled in the embryo until the end of neural tube closure [43].

Our findings link glucose metabolism and cell proliferation within the developing neural tube and may provide insight into how the developing embryo may respond to hyperglycemia if an increased amount of glucose is metabolized leading to an increase in cell proliferation and NTDs. In *miR-302* knockout embryos, we find increased expression of genes involved in glucose import/phosphorylation and the upper glycolysis pathway. Importantly, enzymes of the upper glycolysis pathway have previously been shown to play a key role in cell migration, differentiation, and proliferation [69,79,80,81,82,83]. Notably, the enzymes that catalyze upper glycolysis pathway reactions and glycolytic flux are similarly upregulated upon neoplastic transformation and within proliferative stem cells [41,72,83,84,85,86,87,88,89].

We show that during neural tube closure, predicted *miR-302* targets *Pfkp*, *Pfkfb3*, and *Hk1* are dramatically upregulated upon loss of *miR-302. Pfkp* and *Pfkfb* have not been directly associated with NTDs previously. However, polymorphisms within *Hk1* have been linked to human spina bifida in Chile [90]. The phosphofructokinase gene product (*Pfk*) is the rate-limiting enzyme in glycolysis [65,91]. In cancer cell lines and primary tumors, *Pfkp* is the predominant isoform of *Pfk*, which promotes proliferation, and is the upregulated isoform identified in our analysis [65,67,91,92]. *Pfkfb3* is required for positive and negative feedback of ATP on glycolytic flux and controls the cellular concentration of fructose 2,6-bisphosphate. Thus, our finding of *Pfkfb3* upregulation could disrupt feedback mechanisms regulating glycolysis [68,93,94]. *Pfk* is allosterically activated by fructose 2,6-bisphosphate, and this overrides the inhibitory influence of ATP on *Pfk*, promoting overall glycolytic flux [95,96,97]. Our data suggest that the negative regulation of ATP on glycolytic flux may be lost in *miR-302* knockout embryos due to the selective upregulation of *Pfkfb3*. Importantly, *Pfkfb3* is also known to couple metabolism and proliferation through regulation of cyclin-dependent kinases [35,66,72,84,98,99]. Nuclear localization of fructose-2,6-bisphosphate, the product of PFKFB3, increases the expression and activity of cyclin-dependent kinase-1 to promote G2/M phase progression [72], and we find increased co-expression of *Cdk1* and *Pfkfb3* upon *miR-302* deletion. PFKFB3 also promotes G1 to S phase transition through direct binding to CDK4, thus inhibiting CDK4 degradation [35]. Furthermore, PFKFB3 interacts with CDK4 to stabilize the CDK4–CDC37–HSP90 complex, which promotes the stabilization of kinases including AKT and ERBB2, both of which promote cell growth [35,100,101]. found more cells co-express *Pfkfb3* and *Cdk4* in *miR-302* knockout ectoderm, consistent with increased proliferation.

Our findings link glycolysis with ectodermal cell proliferation, processes that must be coordinately regulated for proper neural tube closure. Since *miR-302* limits glycolytic flux by targeting key genes that encode upper glycolysis pathway enzymes, it is interesting to speculate that the amount of glucose processed by the fetus could be modulated by *miR-302*. Our findings may serve as a platform to study the link between glycolysis and NTDs in other mouse models including diabetes-induced maternal hyperglycemia.

## 4. Materials and Methods

### 4.1. Single-Cell and Library Preparation and Sequencing

*MiR-302-GFP* heterozygous mice were intercrossed, wildtype embryos were harvested at E8.25 (3 somite stage), and wildtype/*miR-302* knockout embryos were harvested at E9.5 (24 somite stage). The cranial region was enzymatically dissociated with papain at room temperature combined with gentle pipetting until a single-cell suspension was achieved. An equal volume of fetal bovine serum (FBS) was used to quench the enzyme. The single-cell suspension was filtered, spun at 300× *g* for 5min, and resuspended in DMEM with 10% FBS. Cell counts were obtained with a hemocytometer and cell viability was assessed using 0.4% Trypan Blue (Thermo-Fisher, Waltham, MA, USA) and was greater than or equal to 98%. GEM generation was performed on a 10× Genomics Chromium Controller Instrument on Chromium Next GEM Chip G (10× Genomics, Pleasanton, CA, USA), and libraries were subsequently prepped using a Chromium Next GEM Single-cell 3′ GEM, Library and Gel Bead Kit v3.1. A Chromium i7 Multiplex Kit was used to index samples, and the resulting libraries were validated using the High Sensitivity NGS Fragment Analysis Kit (Agilent Technologies, Santa Clara, CA, USA) on a 12-Capillary Fragment Analyzer and quantified using the Quant-iT dsDNA Assay Kit (high sensitivity) (Invitrogen, Carlsbad, CA, USA). In total, 2 nM of libraries was subjected to paired-end sequencing on an Illumina NextSeq500 (Illumina, San Diego, CA, USA). From the cranial region 5693 single-cells were sequenced at E8.25 (63,542,136 total reads; average of 11,161 reads per cell) and at E9.5 16,949, cells were sequenced from a wildtype embryo (128,243,041 reads; average of 7566 reads per cell) and 29,386 cells from a staged matched *miR-302* knockout embryo (152,312,550 reads; average of 5183 reads per cell).

### 4.2. Bioinformatic Analysis

Raw bcl files were downloaded using Illumina’s BaseSpaceCLI version 0.10.7 and were converted to fastq files using 10× Genomics Cell Ranger mkfastq version 3.0.2 (10× Genomics, Pleasanton, CA, USA). The resulting fastq files were aligned to the mm10 genome using 10× Genomics Cell Ranger count with mapping rates between 86% and 89%, and we detected between approximately 17,000–19,000 total genes in each sample. Data were processed and visualized using Seurat version 3.1.5. Mitochondrial genes and cell doublets were filtered out. Clustering was performed in Seurat [102] using 4–5 statistically significant principal components based on of their standard deviation. Cells of low complexity with less than 200 UMIs and doublets/triplets with greater than 2500 UMIs were filtered out. Data were log normalized and were integrated for *miR-302* knockout versus wildtype comparison. Visualization of the cells was performed using Uniform Manifold Approximation and Projection for Dimension Reduction (UMAP) algorithm as implemented by Seurat. Cell clusters were identified as previously described based on expression of known markers [55] (Appendix A). Differential gene expression analyses were performed using Seurat’s FindMarkers function, and Gene Ontology (GO) was performed using Enrichr’s Molecular Function and Biological Process GO analyses. Gene expression analyses were performed using the GO gene sets from the Broad Institute molecular signatures database [103]. Each gene list contained the following number of genes: fatty acid metabolism, 63; antioxidant activity, 86; oxidative phosphorylation, 144; glycolysis, 67; folic acid metabolism, 38. The average expression for each gene within each pathway was calculated for each cell population for wild type embryos. For the *miR-302* knockout fold change calculation, average expression for each gene within each pathway was first calculated for both wild type and *miR-302* knockout embryos. The fold change was calculated for each gene individually for each population, and then an average was taken for each cell population and/or pathway. Co-expression analyses were performed on cells where expression of both genes >1. MiRNA targets were identified using TargetScanMouse 7.2 [104]. Cartoons and schematics were produced using Biorender illustrations.

### 4.3. Embryo Processing and Immunofluorescence

Mouse embryos were dissected in phosphate-buffered saline, pH7.4 (PBS), and fixed overnight at 4 °C in 3.7% formaldehyde diluted in PBS. Following fixation, embryos were washed with PBS containing 0.1% Triton (PT), stored in methanol at −30 °C, and rehydrated in PT at the time of use. For cryosectioning, embryos were cryopreserved using a sucrose gradient of 10%, 20%, then 30% *w*/*v* sucrose in PT, followed by 1:1 30% sucrose:OCT Compound (Fisher Scientific, Waltham, MA, USA) and 100% OCT. Embryos were embedded in OCT and flash-frozen in a dry ice and 100% ethanol bath and stored at −80 °C until sectioning. Cryo-sectioning was performed at 10um, and slides were stored at −80 °C until staining. Sections on slides for immunofluorescence were washed with PT and blocked for 1 h at room temperature (RT) in 5% Gibco normal goat serum (Fisher Scientific, Waltham, MA, USA) and 1% bovine serum albumin (Fisher Scientific BP1600100). Primary antibodies were diluted and blocked and applied to tissue overnight at 4 °C (Pfkp 1:100 (Cell Signaling 8164)); Hk1 1:100 (Cell Signaling 2024)). AlexaFluor Secondary antibodies (Thermo-Fisher, Waltham, MA, USA) in blocking buffer were applied for 1.5 h at room temperature. Embryos for wholemount and sections on slides were stained with 1 ug/mL DAPI, and sections were mounted with Fluoromount G (Fisher Scientific OB10001). Images of cross-sections were taken on a Zeiss LSM980 (Zeiss, Oberkochen, Germany), and wholemount embryos were imaged on a Leica M165FC (Leica, Wetzlar, Germany) dissecting microscope with a Leica DFC 3000G camera.

### 4.4. Untargeted Metabolic Profiling

Wildtype and knockout embryos were collected from pregnant dams at E9.5. The heads were dissected in PBS anterior to the otic placode, washed briefly in sterile Milli Q water, snap frozen with liquid nitrogen in a round bottom microcentrifuge tube, and stored at −80 °C until metabolite extraction. The posterior portion of the embryo was reserved for genotyping.

Metabolites from the dissected heads were extracted using an 80:20 methanol:water mixture chilled by dry ice, and the extracts were homogenized by bead-beating for 45 s with a Tissuelyser cell disrupter (Qiagen, Hilden, Germany). The samples were centrifuged at 4 °C for 5 min at 5000 rpm, and the supernatant was reserved at −80 °C, while the extraction procedure was repeated on each debris pellet two additional times. The final debris pellet was reserved for protein quantification using a DC assay (Bio-Rad, Hercules, CA, USA), while the supernatants from each repeated extraction were pooled for each individual sample and the solvent was evaporated in a Vacufuge (Eppendorf, Hamburg, Germany) at 4 °C. The dried extracts were reconstituted in 70% acetonitrile at a relative protein concentration of 1 μg/mL. In total, 5 μL of each sample was injected for LC/MS-based profiling using and an Agilent Model 1290 Infinity II liquid chromatograph coupled to an Agilent 6550 iFunnel time-of-flight mass spectrometer utilizing aqueous normal phase (ANP) chromatography on a Diamond Hydride column (Microsolv, Greater Wilmington, NC, USA). Mobile phases consisted of: (A) 50% isopropanol containing 0.025% acetonitrile and (B) 90% acetonitrile containing 5 mM ammonium acetate. In total, 6μM EDTA was added to the mobile phase to eliminate metal ion interference with chromatograph peak integrity and electrospray ionization. The following gradient was applied: 0–1.0 min, 99% B; 1.0–15.0 min, 20% B; 15.0–29.0, 0% B; 29.–37 min, 99% B. Metabolite measurements were normalized using flanking quality control samples prepared from a pool of all samples, which were run every six injections.

The raw data were analyzed using MassHunter Profinder 8.0 and MassProfiler Professional (MPP) 14.9.1 software (Agilent Technologies, Santa Clara, CA, USA). Metabolite structures were identified using an in-house annotated metabolite database created using MassHunter PCDL manager 7.0 (Agilent Technologies) based on monoisotopic neutral masses (<5 ppm mass accuracy) and chromatographic retention times. A molecular formula generator algorithm in MPP was used, based on weighted consideration of monoisotopic mass accuracy, isotope abundance ratios, and spacing between isotope peaks. A tentative compound ID was assigned when PCDL database and MFG scores concurred for a given candidate molecule. Tentatively assigned molecules were verified based on a match of LC retention times and/or MS/MS fragmentation spectra for pure molecule standards contained in the in-house database. Student t-tests were used to identify metabolites of differential abundance in the LC/MS analysis (α = 0.05).

### 4.5. Phosphofructokinase Assay

Embryos were incubated in 50 uL lysis buffer (50 mM Tris-HCl, pH 7.5, 1 mM EDTA, 150 mM NaCl, 1% NP-40, 1 mM DTT, protease inhibitor cocktail) for 30 min on ice to lyse cells. Reaction buffer (50 mM Tris-HCl, pH 7.5, 5 mM MgCl_2_, 5 mM ATP (Sigma, St. Louis, MO, USA #A2383), 0.2 mM NADH (Sigma #N4505), 100 mM KCl, 5 mM Na_2_HPO_4_, 5 mM MgCl_2_, 0.01 AMP (Sigma #01930), 5 mM fructose-6-phosphate (Sigma #F3627), 5 units of triosephosphate isomerase (Sigma #T2507) per mL, 1 unit of aldolase (Sigma #A2714) per mL, and 10 units of glyceraldehyde-3-phosphate dehydrogenase (Sigma #G2267) per mL) was prepared and brought to room temperature. Cell lysate was added to 500 uL reaction buffer, and optical absorbance was measured at 340 nm at room temperature every 1 min for a total of 60 min with a DeNovix DS11 FX+ spectrophotometer (DeNovix, Wilmington, DE, USA). The resulting absorbance values were plotted for each embryo, and linear regression was used to determine a slope of the linear portion of the curve to estimate Phosphofructokinase activity.

### 4.6. Animal Work

All research and animal care procedures were approved by the Baylor College of Medicine Institutional Animal Care and Use Committee and housed in the Association for Assessment and Accreditation of Laboratory Animal Care-approved animal facility at Baylor College of Medicine under protocol number AN-7033 approved on 30 January 2019. All strains were maintained on C57BL6 background. For adult mouse genotyping, 1–2 mm ear clips were obtained and lysed using 75 uL 25 mM NaOH 0.2 mM EDTA at 98 °C for 1 h and neutralized with 75 uL 40 mM Tris-Cl, pH 5.5. For embryo genotyping, DNA was isolated from yolk sacs and digested overnight in lysis buffer (50 mM Tris-HCl (pH 8.0), 10 mM EDTA, 100 mM NaCl, 0.1% SDS, and 5mg/mL proteinase K). Cell debris was removed, and an equal amount of isopropanol was used to precipitate DNA at −30 °C for 1 h. DNA was isolated by a 30min centrifugation, and a DNA pellet was resuspended in water. PCR was performed using 40 cycles 95 °C for 20 s, and touch-down annealing at 64, 62, 60, 58, and 72 °C for 40 s. All PCR primers and expected band sizes are in Table 1.

## Figures and Tables

**Figure 1 ijms-21-07534-f001:**
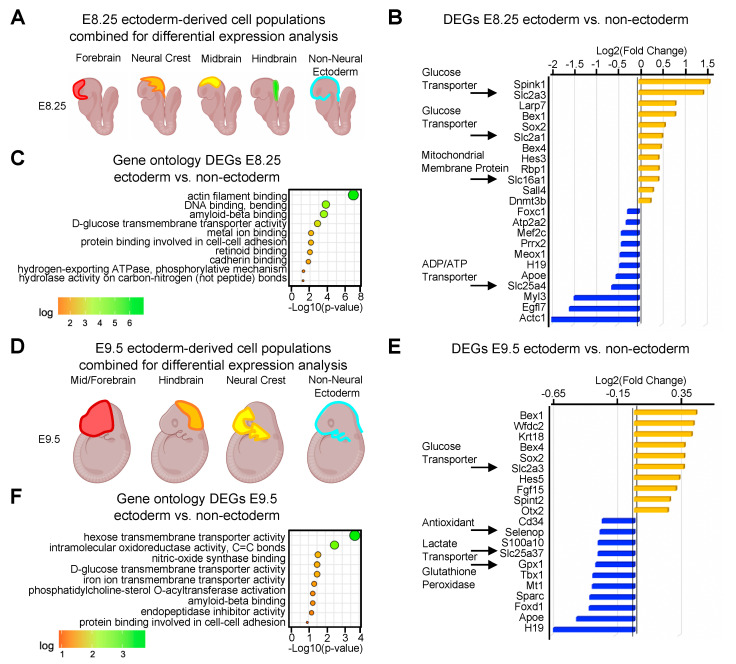
Metabolism and differentiation are coupled during neurulation. (**A**) Differential expression of ectoderm derived cell populations grouped together and compared to the non-ectoderm-derived cells at E8.25. (**B**) Bar plot showing the top up- and downregulated genes at E8.25. Arrows point out metabolic genes. (**C**) Dot plot showing molecular gene ontology of differentially expressed genes at E8.25. (**D**) Differential expression of ectoderm-derived cell populations grouped together and compared to the non-ectoderm-derived cells at E9.5. (**E**) Bar plot showing top up- and downregulated genes at E9.5. Arrows point out metabolic genes. (**F**) Dot plot showing molecular gene ontology analysis of differentially expressed genes at E9.5.

**Figure 2 ijms-21-07534-f002:**
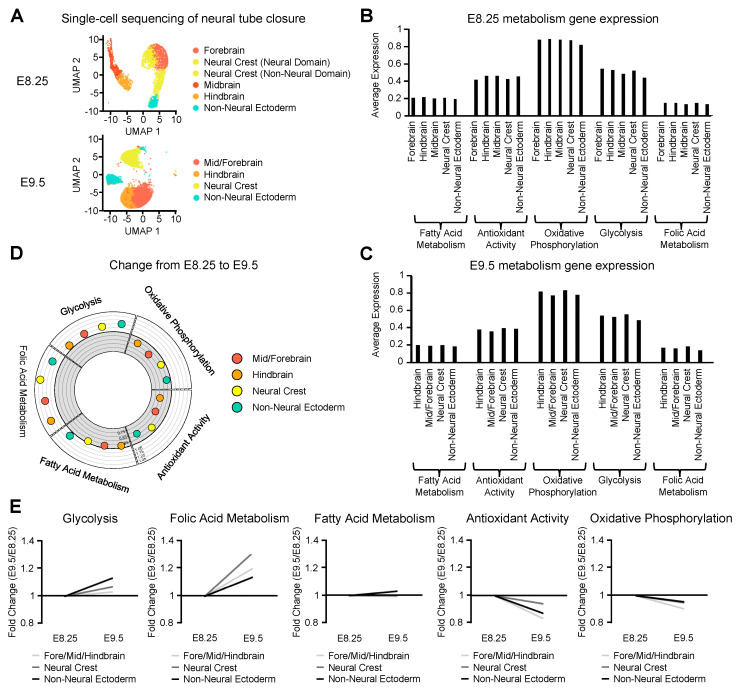
Single-cell sequencing reveals metabolic contributions to neural tube closure. (**A**) Uniform Manifold Approximation and Projection for Dimension Reduction (UMAP) plot showing the ectoderm-derived cell populations that contribute to neural tube closure obtained from single-cell sequencing of the cranial region at E8.25 and E9.5. (**B**) Bar plot showing gene expression of metabolic pathways in ectoderm-derived cell at E8.25 and (**C**) E9.5. (**D**) Circle plot showing the change in expression of metabolic pathways from E8.25 to E9.5 for each ectoderm-derived cell population. The white exterior region of the plot represents upregulation while the grey interior represents downregulation. (**E**) Line graphs showing change in expression for each metabolic pathway during neural tube closure in the neural tube, neural crest, and non-neural ectoderm.

**Figure 3 ijms-21-07534-f003:**
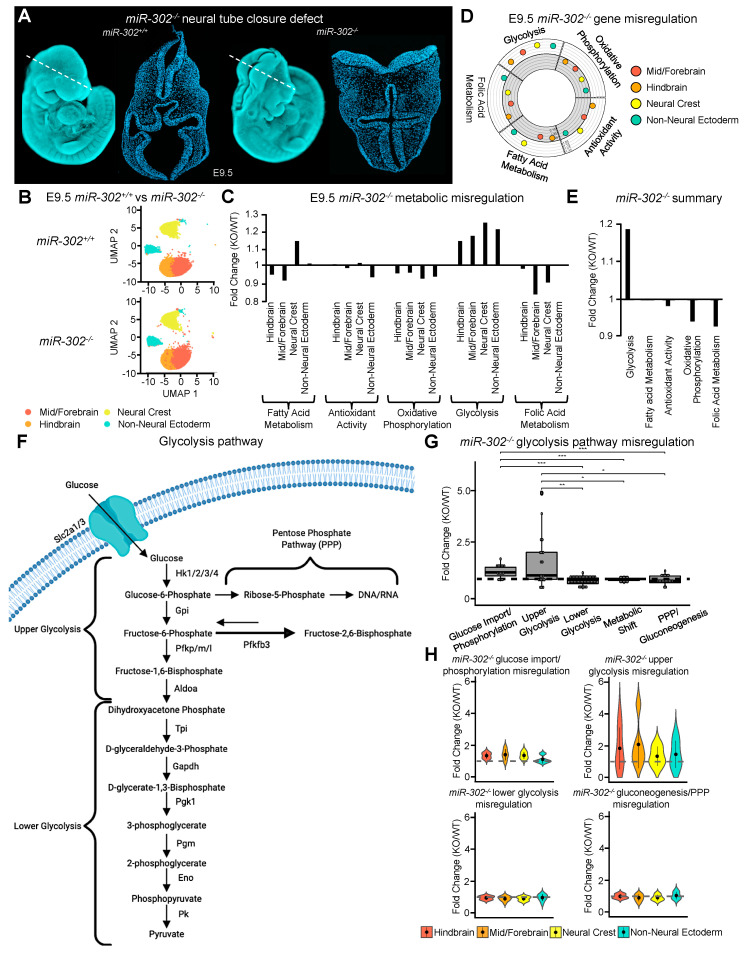
Upper glycolysis is regulated by *miR-302*. (**A**) DAPI staining of wildtype and *miR-302* knockout embryos in both wholemount and transverse cross section to show the resulting cranial neural tube closure defect. (**B**) UMAP plot comparing ectoderm-derived cell of wildtype and *miR-302* knockout embryos at E9.5. (**C**) Bar plot and (**D**) circle plot showing the misregulation of metabolic pathways in ectoderm-derived populations upon *miR-302* deletion. (**E**) Bar plot summarizing the misregulation of genes in each metabolic process upon *miR-302* deletion. (**F**) Schematic showing the process of glycolysis including glucose import, the pentose phosphate pathway, and upper and lower glycolysis. (**G**) Boxplot of the misregulation of the glycolysis pathway of the *miR-302* knockout. (**H**) Violin plots to show population specific misregulation of the glycolysis pathway in the *miR-302* knockout.

**Figure 4 ijms-21-07534-f004:**
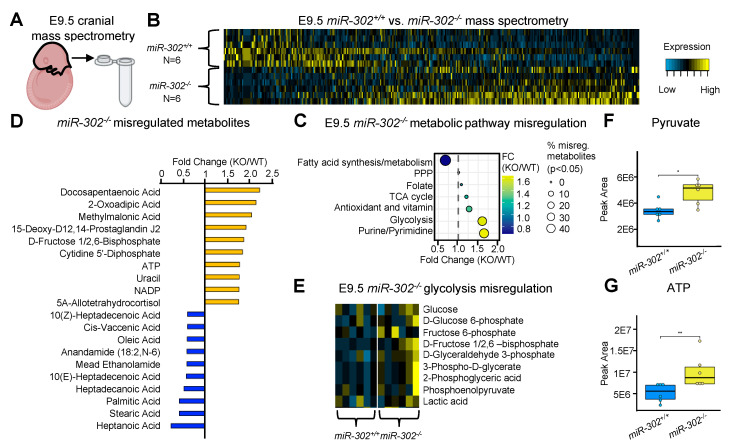
Loss of *miR-302* leads to accumulation of glycolytic intermediates. (**A**) Schematic showing sample collection for mass spectrometry. (**B**) Heatmap of global misregulation of metabolites upon *miR-302* deletion. (**C**) Dot plot showing how metabolic pathways were changed in the *miR-302* knockout. (**D**) Bar plot of the top significantly up- and downregulated metabolites. (**E**) Heatmap showing the upregulation of the glycolysis pathway. (**F**) Boxplot showing upregulation of pyruvate and (**G**) ATP in the *miR-302* knockout.

**Figure 5 ijms-21-07534-f005:**
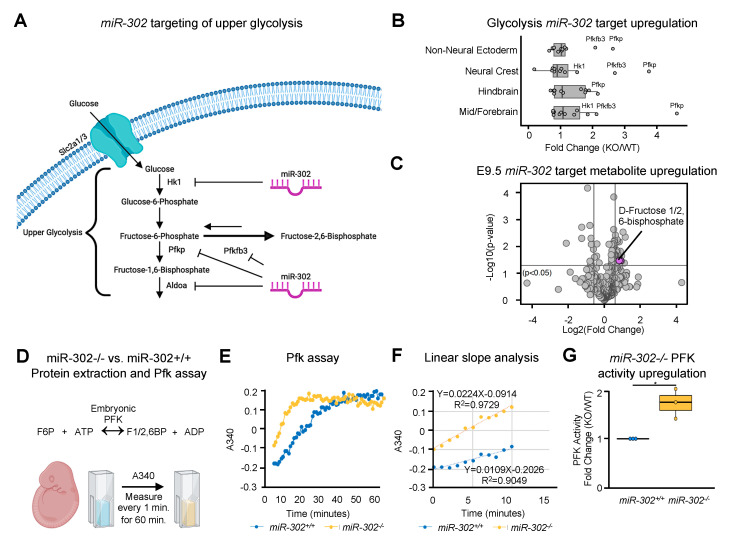
*miR-302* targets *Pfkp*, *Pfkfb3*, and *Hk1* to regulate upper glycolysis. (**A**) Schematic showing *miR-302* targeting of the upper glycolysis steps that control the rate of glycolytic flux. (**B**) Box plot of misregulated *miR-302* targets within the glycolysis pathway. Each *miR-302* target is represented as a data point. (**C**) Volcano plot showing the only significantly upregulated glycolysis metabolite with a fold change >1.5 is that produced by upregulated *miR-302* target *Pfkfb3.* (**D**) Schematic showing protein extraction and PFK assay to measure PFK activity differences between wildtype and *miR-302* knockout embryos. (**E**) Line graph of representative replicate from PFK assay showing change in absorbance over time. (**F**) Linear region of the curve plotted for representative replicate to show slope calculation to represent PFK activity. (**G**) Fold change of PFK activity between wildtype and *miR-302* knockout embryos.

**Figure 6 ijms-21-07534-f006:**
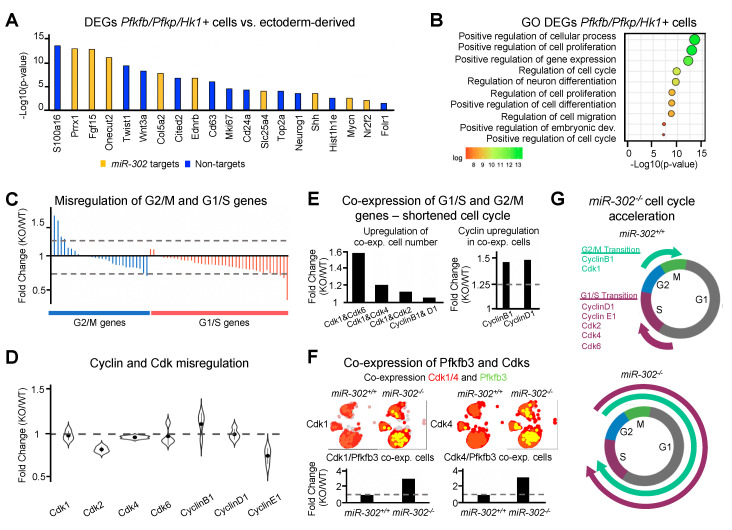
*miR-302* targets *Pfkp*, *Pfkfb3*, and *Hk1* to regulate cell proliferation. (**A**) Representative differentially expressed genes and (**B**) biological gene ontology analysis on the differentially expressed genes obtained from comparing the cells within ectoderm-derived populations expressing *Pfkp*, *Pfkfb3*, or *Hk1* to ectoderm-derived and non-ectoderm-derived cells at E9.5. (**C**) Bar plot of misregulation of genes upon *miR-302* deletion that promote the G2/M and G1/S phase transitions. (**D**) Violin plot of misregulation of Cyclin and Cdk genes of the *miR-302* knockout. (**E**) Bar plots showing number of cells and cyclin expression in cells that co-express G1/S and G2/M genes. (**F**) Blended UMAP plot and bar plot to show number of cells co-expressing *Pfkpfb3* with both *Cdk1* and *Cdk4*. (**G**) Schematic of co-expression of cell cycle transition genes and increased proliferation of the *miR-302* knockout.

**Table 1 ijms-21-07534-t001:** PCR primers and expected band sizes.

Genotyping Primer	Sequence	Expected Band Sizes
*miR-302* GFP forward	CAGGACCTACTTTCCCCAGAGCTG	274 bp wildtype and 547 bp GFP mutant.
*miR-302* GFP wildtype reverse	GAACCCACCCACAAGGCAACTAG	
*miR-302* GFP mutant reverse	GAAGATGGTGCGCTCCTGGACGTAGC

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
