# Peer review of "MiR-302 Regulates Glycolysis to Control Cell-Cycle during Neural Tube Closure"

_ijms, 2020, doi:10.3390/ijms21207534_

Round 1

Reviewer 1 Report

In this manuscript Keuls et al. conducted single cell RNA-seq and mass spectrometry analyses and determined metabolic alterations with increased glycolysis and decreased lipid metabolites in miR-302-null embryos that are known to cause cranial neural tube defects. The upregulation of the predicted miR-302 targets Pfkp, Pfkb3, and Hk1 may be connected with a shortened cell cycle and increased proliferation. They conclude that miR-302 plays a critical role in coordinating the metabolic landscape of neural tube closure. Although the metabolic profiles and major alterations determined in the miR-302 embryonic heads may provide new information to the field, this manuscript has several major issues as detailed below.

Major concerns

  1. This manuscript has a fundamental flaw in research design. All of the major analyses with mutants in this study were carried out at E9.5, after cranial neural tube closed in the wild-type embryos and cranial neural tube defects occurred in the mutant embryos. The major findings in this study may indicate that the metabolic alterations in the ectodermal cells are caused by secondary damage from open brains, but they provide no evidence for cranial neural tube defects that is caused by altered metabolisms or increased glycolysis.
  2. The key findings of altered gene expression in this study are not validated by in situ hybridization and real-time RT-PCR etc.
  3. This manuscript is presented in a way of overstating the conclusions and significance of their own works and ignoring other major published works in the fields. For instance, the title is “MicroRNA regulation of oxidative stress, proliferation, and neural tube closure”, but this study is not directly addressing oxidative stress. The authors state in the abstract that “Dysregulation of glucose metabolism caused by maternal hyperglycemia during pregnancy has been associated neural tube closure defects (NTDs) in humans, however, the underlying mechanism remains unknown”. There is a whole field out there to study oxidative stress, metabolic alterations, and NTD mechanisms caused by maternal hyperglycemia or diabetes, including the involvement of microRNAs in the diabetic embryopathies including NTDs, evident by numerous publications, as listed a few below.

Antioxidative treatment of pregnant diabetic rats diminishes embryonic dysmorphogenesis. Cederberg J, Eriksson UJ. Birth Defects Res A Clin Mol Teratol. 2005 Jul;73(7):498-505. doi: 10.1002/bdra.20144. PMID: 15959875

Developmental damage, increased lipid peroxidation, diminished cyclooxygenase-2 gene expression, and lowered prostaglandin E2 levels in rat embryos exposed to a diabetic environment. Wentzel P, Welsh N, Eriksson UJ. Diabetes. 1999 Apr;48(4):813-20. doi: 10.2337/diabetes.48.4.813. PMID: 10102698

Activation of the hexosamine pathway causes oxidative stress and abnormal embryo gene expression: involvement in diabetic teratogenesis. Horal M, Zhang Z, Stanton R, Virkamäki A, Loeken MR. Birth Defects Res A Clin Mol Teratol. 2004 Aug;70(8):519-27. doi: 10.1002/bdra.20056. PMID: 15329829

Oxidative stress-induced miR-27a targets the redox gene nuclear factor erythroid 2-related factor 2 in diabetic embryopathy. Zhao Y, Dong D, Reece EA, Wang AR, Yang P. Am J Obstet Gynecol. 2018 Jan;218(1):136.e1-136.e10. doi: 10.1016/j.ajog.2017.10.040. Epub 2017 Nov 1. PMID: 29100869 Free PMC article.

Protein kinase C-alpha suppresses autophagy and induces neural tube defects via miR-129-2 in diabetic pregnancy. Wang F, Xu C, Reece EA, Li X, Wu Y, Harman C, Yu J, Dong D, Wang C, Yang P, Zhong J, Yang P. Nat Commun. 2017 May 5;8:15182. doi: 10.1038/ncomms15182. PMID: 28474670 Free PMC article.

  1. 1B-C: The authors should indicate the number of genes that is involved for each GO enrichment pathway since the average expression was calculated.
  2. Page 3, line 97-100: It would make more sense if authors could use normal GO enrichment analysis and P value to indicate Glycolysis is the most significant. Same for Fig. 1E, Fig. 3C, E
  3. Page 5-6, line 171-173: It is unclear why there is a link between glucose metabolism and differentiation based on only gene ontology results.
  4. 3G,H: The authors need to provide more details on how to calculate the fold change and how many genes involved in each pathway. It is inappropriate to calculate fold change if more than one gene enriched in a specific pathway.
  5. Page 9, line 240-242: The authors mentioned “predicted targets”, which needs to provide what kind of method has been used on prediction of the miR-302 targets. The total prediction list should be provided and the validation for these 4 targets should be proved since three of them induce the final conclusion, otherwise it seems very speculative without much more details.
  6. Page 11, line 279: The authors need to define “Upregulated”. Is it base on fold change, Padj value, or any other standards?
  7. Page 12, line 328, 329: It is overstating that “Our findings …may provide insight into how hyperglycemia leads to birth defects”. Birth defects or NTDs caused by hyperglycemia are not addressed in this study. The observed increase of several genes in glycolysis has nothing to do with hyperglycemia unless the authors can prove it.

Minor concerns:

Fig. 1A,D: inconsistent coloring for neural crest cells and hindbrain regions between E8.25 and E9.5.

Fig. 1B,C,E: The gene lists of metabolic pathways from single cell RNA-seq should be provided as supplementary tables.

Fig. 1E: “Brain” is more accurate than “Neural Tube” if the forebrain, midbrain, and hindbrain regions are combined.

The panel orders in Figs. 1 and 2 are not linear.

Page 4, line 123-125: References are need for Hif1a and Nrf1. Figure 1F is cited in the text, but not in the figure 1.

Fig. 2C,2F: All DEG lists for the gene ontology in these figure panels should be provided as supplementary tables.

Fig. 3. The DEG lists in various metabolic pathways should be provided as supplementary tables.

Why is it sometimes “Migratory Neural Crest Cells” vs. “Neural Crest Cells” when referring to the same timepoint? i.e. Figure 3B vs. 3D

Page 9, line 239 onwards: any of these predicted miR-302 targets are associated with cranial NTDs? It is important to include the details and validate these key candidate targets.

Page 11, line 292: The authors need to define co-expression in this paper.

The way of supplemental figure labeling is confusing, for instance, Figure 1 – Supplement 1; Figure 2 – Supplement 1. It would be better to label as Supplement Figure S1, S2.

Also, the figure legends are absent for the Supplementary Data with only some sections explained in the main text.

It might be better to keep the same order of cell populations/brain regions in the panels of Supplemental Figure 1. ROS in C is not defined.

Reviewer 2 Report

The manuscript entitled "MicroRNA Regulation of Oxidative Stress, Proliferation, and Neural Tube Closure" by Keuls et al., describes the role of miR-302 in the regulation of upper glycolysis pathway carbon flux and its association with proliferation processes of the ectoderm during neural tube closure. The authors used specific methodologies such as single-cell transcriptomic analysis and mass-spectrometry to indentify metabolic profile comparisons between successful and failed neurulation in mouse.

The study is highly interesting providing significant results on the metabolic contribution to neural tube closure such as the link between glucose metabolism and cell proliferation within the developing neural tube. Both experimental data and bioinformatic analysis during neurulation are clear and well organized. The manuscript is well written and coherent and can be published.

As a minor comment, the addition of m/z values and mass spectra of metabolites can also be included.

Reviewer 3 Report

Overall, the authors presented a nice link between fetal hyperglycemia and neural tube closure defects primarily using the single cell RNAseq technique. I have the following three concerns.

  1. 3c, the authors showed a fold change distribution of miR-302-/- different genes, it would be helpful for the readers to see as such average gene expression in addition to fold change comparison.
  2. Secondly, to extent the RNAseq data, the authors could present protein level data of miR-302 target genes Pfkp, Pfkfb3, and Hk1 at E9.5 through immunostaining (semi-quantification) or western blot ( with quantification).
  3. It would be helpful for the readers to follow-up the work if the authors could upload the whole single cell gene expression data of control and miR-302-/- at the NCBI database (omnibus data).

Round 2

Reviewer 1 Report

The revised manuscript has been improved, but several issues remain and should be solved before publication.

  1. Regardless the strain differences, the E9.5 mouse embryos used in the study clearly completed cranial neural tube closure, which should be reflected in the title and the text.
  2. The major analyses of single cell RNA-seq and mass spectrometry in this study were conducted at E9.5 whole heads, not restricted to neural tube closure or the entire embryos. Their alterations could be secondary effects after cranial neural tube failed to close in the mutants, which should be reflected in the title and the text.
  3. The miR-302 targets and their roles in cell cycle progression and neural tube closure are not demonstrated in this study, which should be clearly stated in the manuscript. Their cell lineage-specific roles also should be interpreted. 

Reviewer 3 Report

I wish you all the best to the authors